

# Neighborhood-based inference and restricted Boltzmann machine for small molecule-miRNA associations prediction

Jia Qu[1], Zihao Song[1], Xiaolong Cheng[1], Zhibin Jiang[2] and Jie Zhou[2]

[1] School of Computer Science and Artificial Intelligence, Changzhou University, Changzhou, Jiangsu, China
[2] Department of Computer Science and Engineering, Shaoxing University, Shaoxing, Zhejiang, China

## ABSTRACT

**Background**. A growing number of experiments have shown that microRNAs (miRNAs) can be used as target of small molecules (SMs) to regulate gene expression for treating diseases. Therefore, identifying SM-related miRNAs is helpful for the treatment of diseases in the domain of medical investigation.

**Methods**. This article presents a new computational model, called NIRBMSMMA (neighborhood-based inference (NI) and restricted Boltzmann machine (RBM)), which we developed to identify potential small molecule-miRNA associations (NIRBMSMMA). First, grounded on known SM-miRNAs associations, SM similarity and miRNA similarity, NI was used to predict score of an unknown SM-miRNA pair by reckoning the sum of known associations between neighbors of the SM (miRNA) and the miRNA (SM). Second, utilizing a two-layered generative stochastic artificial neural network, RBM was used to predict SM-miRNA association by learning potential probability distribution from known SM-miRNA associations. At last, an ensemble learning model was conducted to combine NI and RBM for identifying potential SM-miRNA associations.

**Results**. Furthermore, we conducted global leave one out cross validation (LOOCV), miRNA-fixed LOOCV, SM-fixed LOOCV and five-fold cross validation to assess performance of NIRBMSMMA based on three datasets. Results showed that NIRBMSMMA obtained areas under the curve (AUC) of 0.9912, 0.9875, 0.8376 and $0.9898 \pm 0.0009$ under global LOOCV, miRNA-fixed LOOCV, SM-fixed LOOCV and five-fold cross validation based on dataset 1, respectively. For dataset 2, the AUCs are 0.8645, 0.8720, 0.7066 and $0.8547 \pm 0.0046$ in turn. For dataset 3, the AUCs are 0.9884, 0.9802, 0.8239 and $0.9870 \pm 0.0015$ in turn. Also, we conducted case studies to further assess the predictive performance of NIRBMSMMA. These results illustrated the proposed model is a useful tool in predicting potential SM-miRNA associations.

## INTRODUCTION

MicroRNAs (miRNAs) are a type of short noncoding RNA molecules existing in viruses, plants and animals, which can regulate gene expression by targeting messenger RNAs (mRNAs) (*Bartel, 2004*; *Zhang, Wang & Pan, 2007*). In 1993, the first miRNA lin-4 was

Corresponding author
Jie Zhou, sxuj_zhou@163.com

identified from the Caenorhabditis elegans (*Lee, Feinbaum & Ambros, 1993*). In 2000, the first human miRNA let-7 was discovered by research workers (*Saliminejad et al., 2019*). Along with an increase in the number of miRNA studies, research shows that miRNAs play a key role in multiple biological processes such as stem cell differentiation, organ development, immune response and insulin secretion (*Cuellar & McManus, 2005*; *Krützfeldt et al., 2005*; *Zhang, Wang & Pan, 2007*). Moreover, some studies showed that abnormal expression of miRNAs is closely linked to some diseases (*Huang, 2017*; *Tili et al., 2008*; *Trang, Weidhaas & Slack, 2008*). For example, overexpression of miR-15a can cause cell apoptosis and further lead to rheumatic diseases (*Ceribelli et al., 2011*). Furthermore, research indicates that low expression levels of miR-1 and miR-13 can cause cardiac hypertrophy (*Carè et al., 2007*). In lung cancer tissues, miR-17-92 clusters, containing miR106a, miR17-5p and miR19a, usually have high expression levels (*Hayashita et al., 2005*). An accumulating body of research indicates that abnormal expression of miRNAs can cause diseases, miRNAs can considered as potential therapeutic targets for treating diseases (*Fasanaro et al., 2010*).

Small molecules (SMs) are a kind of organic compounds with low molecular weight (*Qu et al., 2018*). Over the past two decades, a number of studies have indicated that small molecules can regulate miRNA expression and inhibit growth and chemoresistance of disease cells (*Liu et al., 2013*; *Zhang et al., 2010*). For example, miR-200b is up-regulated in various tumors, which can be down-regulated by using SM of 5-fluorouracil (*Rossi, Bonmassar & Faraoni, 2007*). Furthermore, streptomycin can target precursor of miR-21 for inhibiting miR-21 expression and treat breast cancer and lung cancer (*Bose et al., 2012*). Also, it was previously reported that 5-aza-2′-deoxycytidine can increase expression of miR-148a by testing with quantitative real-time polymerase chain reaction, which has positive implications for the treatment of gastric cancer (*Sun et al., 2014*). Unsurprisingly, there is a close association between SMs and miRNAs. Therefore, identifying potential SM-miRNA associations can contribute to the treatment of diseases. However, traditional experimental approaches are time-consuming and labor-intensive in identifying new SM-miRNA associations. The development of innovative computational techniques for identifying potential SM-miRNA associations is an urgent necessity.

In recent years, many gene expression-based computational models have been developed to predict potential SM-miRNA associations. For example, *Wang et al. (2016)* developed a computational model of functional similarity to predict potential SM-miRNA associations. First, they collected gene expression profiles of 88 miRNAs from the Gene Expression Omnibus (GEO) database and collected gene expression profiles of 1,309 SMs from Connectivity Map (CMAP) database. Then, differentially expressed genes of all SM-miRNA pairs can be obtained based on significance analysis of microarray (SAM). According to differentially expressed genes between SMs and miRNAs, they computed functional similarity scores of all SM-miRNA pairs by employing Gene Ontology (GO) enrichment analysis. According to transcriptional responses after miRNA regulation and drug therapy, *Jiang et al. (2012)* presented a model to predict potential SM-miRNA associations in 23 different human cancers by utilizing a Kolmogorov–Smirnov (KS) test based on differentially expressed miRNA target genes and gene signatures obtained from gene
expression profile of SMs treatment. Based on predicted SM-miRNA associations, they built SM-miRNA networks (SMirNs) for 17 cancers. In each cancer of SMirN, they identified miRNA modules in which a group of miRNAs are linked to the same SM and further identified SM modules in which SMs are relate to the same miRNA. Subsequently, based on SMirNs and identified modules, they utilized structural similarity of small molecules and assessed miRNAs with miRNA family, targets or GO functional similarities to predict new drug target miRNAs and drug candidates for cancer therapy. Particularly, this study was expanded by *Meng et al. (2014)* to identify SMs for Alzheimer's disease (AD)-related miRNAs (ADMs). The study utilized the differentially expressed target genes of ADMs and gene expression signatures of bioactive SM perturbations to predict potential SM-ADM associations using KS scores. Then, they built network of SM-miRNA association in AD (SmiRN-AD) based on predicted SM-ADM associations and proceeded to conduct a more detailed analysis of the topological and functional properties of the SmiRN-AD network.

Furthermore, *Lv et al. (2015)* presented a network-based approach for identifying potential SM-miRNA associations. They constructed an integrated SM-miRNA association network by employing SM similarity network, miRNA similarity network and known SM-miRNA associations network. Then, a similarity-based random walk with restart (RWR) algorithm was used on the constructed network for predicting potential associations between SMs and miRNAs. *Li et al. (2016)* also developed a network-based model, namely small molecule-miRNA network-based inference (SMiR-NBI), to identify new SM-miRNA associations. First, heterogeneous network was built by linking drugs, miRNAs and genes. Second, the initial score of a SM were preset and distributed averagely to its associated miRNAs. Then, the initial scores of miRNAs were distributed averagely to its associated SMs and SMs redistribute received scores from miRNAs for every associated miRNA. Finally, end scores between SMs and miRNAs were obtained to represent the likelihood of associations. In addition, *Qu et al. (2018)* developed a new computational model of triple layer heterogeneous network for identifying potential SM-MiRNA associations (TLHNSMMA), in which they used disease information as a bridge to construct a triple layer heterogeneous network based on known SM-miRNA associations, known miRNA-disease associations, SM similarity, miRNA similarity and disease similarity. Then, an iterative updating algorithm was used on the triple layer heterogeneous network for predicting potential SM-miRNA associations. *Qu et al. (2019)* also developed a computational model of HeteSim-based inference for predicting new SM-MiRNA associations (HSSMMA) where a heterogeneous network was constructed based on multiple biological data. After that, HeteSim, a path-constrained measurement method, was used on the constructed heterogeneous network to compute paths between an SM and a miRNA for predicting potential SM-miRNA associations. Recently, *Chen et al. (2021)* constructed a computational model of bounded nuclear norm regularization (BNNR) to predict new SM-miRNA associations (BNNRSMMA). First, a SM-miRNA association heterogeneous network was built by integrating miRNA similarity, SM similarity and known SM-miRNA associations. Then, a target matrix was defined to represent the heterogeneous network. Moreover, the target matrix was completed by minimizing its

nuclear norm with alternating direction method of multipliers (ADMM) and potential SM-miRNA associations as a part of completed target matrix were obtained.

Based on previous studies, some single predictors usually have poor generalization performance in identification of potential SM-miRNA associations. Combining two weak predictors by ensemble learning can improve model generalization performance. Moreover, some network-based models cannot be applied to predict potential miRNAs (SMs) for new SM (miRNA) without any known associated miRNA (SM). Therefore, we developed a novel computational model of neighborhood-based inference (NI) and restricted Boltzmann machine (RBM) for identifying potential small molecule-miRNA associations (NIRBMSMMA) by integrating network-based model of NI and machine learning-based mode of RBM. In the model, for a specific SM and a specific miRNA, NI was carried out to predict potential association between the SM and the miRNA by reckoning the sum of know associations between the miRNA (SM) and neighbors of the SM (miRNA) based on SM (miRNA) similarity. Then, a two-layer network based model of RBM was employed to predict potential SM-miRNA associations by fully learning probability distribution governing known SM-miRNA associations. At last, an ensemble learning model was used to integrated NI and RBM for obtaining potential SM-miRNA associations. To evaluate predictive ability of NIRBMSMMA, we implemented global leave one out cross validation (LOOCV), miRNA-fixed LOOCV, SM-fixed LOOCV and five-fold cross validation based on three different SM-miRNA association datasets. Furthermore, we conducted two types of case studies for assessing the performance of NIRBMSMMA. Result of the first case study showed that five out of the top 20 and 10 out of the top 50 predicted SM-miRNA associations were verified by searching published literatures on PubMed. In the second case study, 5-fluorouracil and gemcitabine were selected as investigated SM respectively. Potential miRNAs related to investigated SM would be identified by removing all known associations between miRNA and the SM. As a result, 32 out of the top 50 predicted miRNAs for 5-fluorouracil and 32 out of the top 50 predicted miRNAs for gemcitabine were validated by SM2miR database and published literature, respectively.

## MATERIALS & METHODS

### SM-miRNA associations

In this article, the known SM-miRNA associations information was obtained from SM2miR v1.0 (*Liu et al., 2013*), and we constructed two different SM-miRNA associations dataset of dataset 1 and dataset 2 (*Jiang et al., 2009*; *Knox et al., 2011*; *Lu et al., 2008*; *Ruepp et al., 2010*; *Wang et al., 2009*). Dataset 1 contains 664 known SM-miRNAs associations between 831 SMs and 541 miRNAs, in which 792 (out of 831, 95%) SMs do not have any known associated miRNAs according to SM2miR. By removing the information of SMs (miRNAs) that have any no known related miRNAs (SMs) in dataset 1, we constructed dataset 2 consists of 39 SMs and 286 miRNAs that are fully involved in the 664 known associations. Unsurprisingly, dataset 2 contains 664 known SMs-miRNAs associations between 39 SMs and 286 miRNAs. Moreover, an adjacency matrix $A \in R^{ns \times nm}$ was built to denote the associations between SMs and miRNAs, where $ns$ represents number of SMs and $nm$

indicates number of miRNAs. If the SM $s(i)$ is associated with miRNA $m(j)$, the entry $A(i,j)$ is equal to 1, otherwise 0.

## Integrated SM similarity

Referring to the earlier study ($Lv\ et\ al.,\ 2015$), the integrated SM similarity was obtained by combining four types of SM similarity including SM chemical structure similarity $S_S^C$ ($Hattori\ et\ al.,\ 2003$), disease phenotype-based similarity for SMs $S_S^D$ ($Gottlieb\ et\ al.,\ 2011$), SM side effect similarity $S_S^S$ ($Gottlieb\ et\ al.,\ 2011$) and gene functional consistency-based similarity $S_S^T$ ($Lv\ et\ al.,\ 2012$). In this study, the SIMilar COMPound (SIMCOMP) approach was utilized to calculate the similarity between SM chemical structures based on SM chemical structure information ($Hattori\ et\ al.,\ 2003$). Then, the Jaccard score was employed, utilizing information about diseases associated with each small molecule to assess the similarity of SMs based on disease phenotypes ($Gottlieb\ et\ al.,\ 2011$). SM side effect similarity was gained with the method of Jaccard score ($Gottlieb\ et\ al.,\ 2011$) based on SM-side effect association dataset. Gene functional consistency-based similarity was derived with Gene Set Functional Similarity (GSFS) ($Lv\ et\ al.,\ 2012$) method on the dataset of SMs' target genes. After that, we used the weighted average approach to calculate the four kind of SM similarity for obtaining integrated SM similarity, which can be described as follows.

$$S_S = \frac{\beta_1 S_S^D + \beta_2 S_S^T + \beta_3 S_S^C + \beta_4 S_S^S}{\sum_{i=1}^{4} \beta_i} \tag{1}$$

where values of $\beta_i (i = 1, 2, 3, 4)$ are set as 1 and $S_S$ denotes integrated SM similarity.

## Integrated miRNA similarity

Similarly, we obtained integrated miRNA similarity by merging disease phenotype-based similarity for miRNAs $S_M^D$ ($Gottlieb\ et\ al.,\ 2011$) with gene functional consistency-based similarity for miRNAs $S_M^T$ ($Lv\ et\ al.,\ 2012$). Specifically, we utilized the GSFS method to determine the functional consistency-based similarity of miRNA target gene sets ($Lv\ et\ al.,\ 2012$). Furthermore, using the Jaccard score, we calculated the disease phenotype-based similarity for miRNAs based on the dataset of miRNA-related diseases ($Gottlieb\ et\ al.,\ 2011$). We employed a weighted average approach to determine the integrated miRNA similarity $SM$, which is outlined below.

$$SM = \frac{\alpha_1 S_M^D + \alpha_2 S_M^T}{\sum_{j=1}^{2} \alpha_j} \tag{2}$$

where values of $\alpha_j (j = 1, 2)$ are set as 1.

## NIRBMSMMA

In this study, we presented a novel computational model, NIRBMSMMA, to identify potential SM-miRNA associations by leveraging known SM-miRNA associations, as well as integrated SM similarity and integrated miRNA similarity. First, we employed NI technique to anticipate potential associations between SMs and miRNAs by varying thresholds to compute the sum associations between the miRNAs (SMs) and neighbors of SM (miRNA).

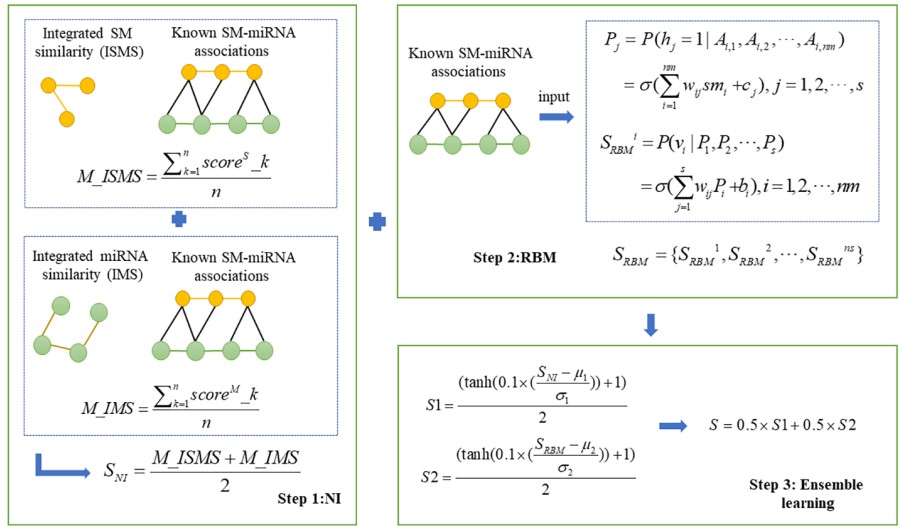

**Figure 1** Flowchart for predicting the potential SM-miRNA associations by using NIRBMSMMA.

Second, a potential SM-miRNA association matrix was predicted using the contrastive divergence algorithm and sigmoid function of the restricted Boltzmann machine (RBM). Considering single method may be weak in generalization ability, ensemble learning was employed to integrate the predictions from NI and RBM to derive the final potential associations between SMs and miRNAs. The flowchart of NIRBMSMMA is shown as Fig. 1.

## Neighborhood-based inference

Neighborhood-based method is a classic collaborative filtering algorithm, also referred to as memory-based algorithm, can recommend similar items for a user based on behaviors of similar users (*Aggarwal, 2016*). Here, we constructed a based model of NI to identify potential SM-miRNA associations. First, we built a NI model utilizing the integrated SM similarity. For a SM $s_{i,i=1,2,...,ns}$, by applying a threshold $\sigma$ based on integrated SM similarity, the neighbors can be filtered, and subsequently, a defined set of neighbors can be established as $\{s_i | ISMS_{d,i} > \sigma, d \neq i\}$. After that, the potential score of SM $s_i$ associated with miRNA $m_j$ can be gained by calculating the sum associations between the miRNA $m_j$ and neighbors of SM $s_i$ as follows.

$$score_{i,j}^S = \frac{\sum_{i=1}^{ns} \sum_{j=1,d\neq i,ISMS_{d,i}\geq\sigma}^{nm} A_{d,j} \times ISMS_{d,i}}{\sum_{i=1}^{ns} \sum_{j=1,d\neq i,ISMS_{d,i}\geq TS}^{nm} ISMS_{d,i}} \tag{3}$$

where $ns$ denotes the count of SM, $nm$ denotes the count of miRNA, $A_{d,j}$ denotes the entry between SM $s_d$ and miRNA $m_j$ in SM-miRNA associations matrix $A$. The $ISMS_{d,i}$ denotes integrated SM similarity between SM $s_d$ and SM $s_i$. The $score_{i,j}^S$ represents potential association score between SM $s_i$ and miRNA $m_j$.

NI model was constructed by filtering neighbors with the threshold $\sigma$. To reduce bias of neighbor selection, we generated multiple thresholds $\{\sigma_1, \sigma_2, ..., \sigma_n\}$ in the range of

0.05 to 0.5 to construct multiple basic NI models. Particularly, an upper bound parameter $\sigma_{upper}$ was employed to determine the multiple thresholds which were represented as $\sigma_{threshold} = \{\sigma_i | \sigma_i \leq \sigma_{upper}, i = 1, 2, \ldots, n(n = |\sigma_{threshold}|)\}$. Thus, $n$ thresholds $\{\sigma_1, \sigma_2, \ldots, \sigma_n\}$ were employed to construct $n$ basic NI models.

After that, the potential score for SM-miRNA associations was derived by integrating $n$ basic NI models as follows.

Then, we integrated $ns$ basic models to predict potential SMs-miRNA associations score by using average strategy, which can be described as follows:

$$M\_ISMS = \frac{\sum_{k=1}^{n} score^{S}\_k}{n} \tag{4}$$

where $n$ denotes the number of basic NI models, $score^{S}\_k$ denotes predicted SM-miRNA associations score based on the $k - th$ threshold and $M\_ISMS$ is potential SM-miRNA associations score based on integrated SM similarity.

Second, we built a NI model based on integrated miRNA similarity by using the method above. For a miRNA $m_{j, j=1,2,\ldots,nm}$, its neighbors can be filtrated by using a threshold $\sigma$ based on integrated miRNA similarity and set of neighbors can be was defined as $M_j | IMS_{t,j} > \sigma, t \neq j$. On the basis of the set of neighbors above, potential association score $score_{i,j}^{M}$ between SM $s_i$ and miRNA $m_j$ was obtained through computing the sum of associations between SM $s_i$ and neighbors of miRNA $m_j$ as Eq. (5).

$$score_{i,j}^{M} = \frac{\sum_{i=1}^{ns} \sum_{j=1, t \neq i, IMS_{t,j} \geq \sigma}^{nm} A_{i,t} \times IMS_{t,j}}{\sum_{i=1}^{ns} \sum_{j=1, t \neq j, IMS_{t,j} \geq \sigma}^{nm} IMS_{t,j}} \tag{5}$$

where $ns$ represents number of SM, $nm$ represents number of miRNA, $A_{i,m}$ denotes the entry between SM $s_i$ and miRNA $m_m$ in SM-miRNA association matrix $A$. The $IMS_{t,j}$ represents integrated miRNA similarity between miRNA $m_t$ and miRNA $m_j$.

To mitigate bias stemming from neighbor selection, we employed several distinct thresholds to construct multiple basic NI models. Next, we combined the multiple basic NI models to generate score for potential SM-miRNA associations, as outlined below.

$$M\_IMS = \frac{\sum_{k=1}^{n} score^{M}\_k}{n} \tag{6}$$

where $n$ denotes the number of basic NI models, $score^{M}\_k$ denotes predicted SM-miRNA associations score based on $k - th$ threshold and $M\_IMS$ denotes potential SM-miRNA associations score based on integrated miRNA similarity.

Finally, the potential SM-miRNA association score $S_{NI}$ was acquired by integrating NI models based on integrated SM similarity and NI models based on integrated miRNA similarity as follows.

$$S_{NI} = \frac{M\_ISMS + M\_IMS}{2} \tag{7}$$
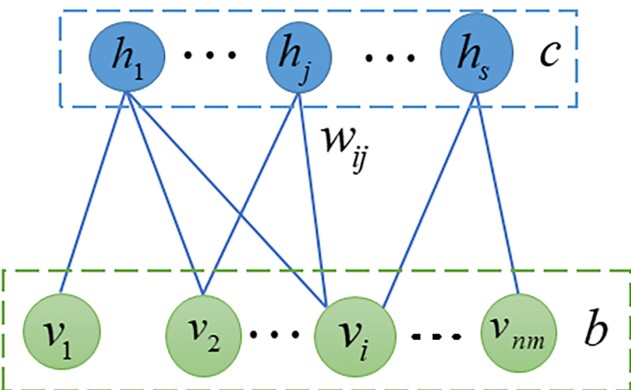

*Hidden layer*

*Visible layer*

**Figure 2 Structure diagram of RBM.**

## Restricted Boltzmann Machine (RBM)

RBM is probabilistic graphical model, also called stochastic neural network, can extract useful features from input data (*Fischer & Igel, 2012*; *Zhang et al., 2018*). In the previous decade, RBM has been extensively studied and employed in many fields including quantum physics (*Melko et al., 2019*), bioinformatics (*Wang & Zeng, 2013*) and text mining (*Monti et al., 2016*). In this article, we used RMB as a based model to predict potential SM-miRNA associations. As shown in Fig. 2, the RBM can be described as a two-layered neural network, consisting of visible layers and hidden layers, each of which is made up of many nodes (units). In an RBM, if we take into account the total number of nodes, there are *nm* in the visible layer and *s* nodes in the hidden layer. We employed $\mathbf{v} = (v_i, v_2, \ldots, v_{nm})$ to represent set of visible layer nodes and used $\mathbf{h} = (h_1, h_2, \ldots, h_s)$ to represent set of hidden layer nodes. As there are no connections within the visible or hidden layer, the energy function between $\mathbf{v}$ and $\mathbf{h}$ can be expressed in the following manner.

$$E(\mathbf{v}, \mathbf{h}) = -\sum_{i=1}^{nm} b_i v_i - \sum_{j=1}^{s} c_j h_j - \sum_{i=1}^{i=nm} \sum_{j=1}^{s} w_{i,j} v_i h_j \tag{8}$$

where *nm* represents number of visible layer nodes, *s* denotes number of visible layer nodes, $b_i$ denotes bias of $i - th$ visible layer node $v_i$, $c_j$ denotes bias of $j - th$ hidden layer node $h_j$ and $w_{ij}$ denotes the weight between $v_i$ and $h_j$.

Based on Eq. (8), the potential association probability of $\mathbf{v}$ and $\mathbf{h}$ can be defined as follows.

$$P(\mathbf{v}, \mathbf{h}) == \frac{1}{Z} e^{-E(\mathbf{v}, \mathbf{h})} \tag{9}$$

where $Z = \sum_{\mathbf{v}, \mathbf{h}} -E(\mathbf{v}, \mathbf{h})$ is called partition function.

Then, we can obtain marginal distribution of visible layer nodes by following equation.

$$P(\mathbf{v}) = \sum_{\mathbf{h}} P(\mathbf{v}, \mathbf{h}) = \frac{1}{Z} \sum_{\mathbf{h}} e^{-E(\mathbf{v}, \mathbf{h})}. \tag{10}$$

Since distributions of nodes of visible or hidden layer are independent, their conditional probabilities can be defined, respectively, as follows.

$$P(v_i = 1 | \mathbf{h}) = \sigma \left( \sum_{j=1}^{s} w_{ij} h_j + b_i \right) \tag{11}$$

$$P(h_j = 1 | \mathbf{v}) = \sigma \left( \sum_{i=1}^{nm} w_{ij} v_i + c_j \right) \tag{12}$$

where $\sigma(x) = 1/(1 + e^{-x})$ is the logistic function.

Based on a dataset containing known SM-miRNA associations between $nm$ miRNAs and $ns$ SMs, an RBM is created for predicting potential SM-miRNA associations, with $nm$ nodes in the visible layer and $s$ nodes in the hidden layer. Each SM is associated with an observation $sm_i = \{A_{i,1}, A_{i,2}, \ldots, A_{i,nm}\}$, which has a binary value indicating whether the SM $s_i$ is associated with $nm$ miRNAs. Based on known SM-miRNA association matrix $A \in R^{ns \times nm}$, we can obtain $ns$ observations with $nm$ dimension. The observation of an SM would be utilized as input for the RBM when predicting new miRNAs for it. Then, the prediction is implemented as two steps below.

$$P_j = P(h_j = 1 | A_{i,1}, A_{i,2}, \ldots, A_{i,nm}) = \sigma \left( \sum_{t=1}^{nm} w_{ij} A_{i,t} + c_j \right), j = 1, 2, \ldots, s \tag{13}$$

$$S_{RBM}^i = P(v_i | P_1, P_2, \ldots, P_s) = \sigma \left( \sum_{j=1}^{s} w_{ij} P_i + b_i \right), i = 1, 2, \ldots, nm \tag{14}$$

where $\sigma(x) = 1/(1 + e^{-x})$ is the logistic function. $S_{RBM}^i$ is predicted score between SM $s_i$ and $nm$ miRNAs. Finally, we obtained the potential SM-miRNA association scores and defined $S_{RBM}$ to save scores as follows:

$$S_{RBM} = S_{RBM}^1, S_{RBM}^2, \ldots, S_{RBM}^{ns} \tag{15}$$

## Ensemble learning

Because single predictor usually has poor generalization performances, ensemble learning is frequently used to combine multiple weak predictors and improve generalization performance. *Dong et al. (2020)*. In the past decades, ensemble learning as one research hot spot have been studied and employed in many fields including data stream classification (*Gomes et al., 2017*), statistical mechanics (*Krogh & Sollich, 1997*), sentiment classification

(*Wang et al., 2014*) and bioinformatics (*Chen, Zhou & Zhao, 2018*). Furthermore, NI and RBM were employed as individual base predictors for ensemble learning to increase the precision of predictions of potential SM-miRNA associations. To ensure the predicted score in range of 0 to 1, we normalized predicted scores of NI and RBM as follows:

$$S1 = \frac{(\tanh(0.1 \times (\frac{S_{NI} - \mu_1}{\sigma_1})) + 1)}{2} \tag{16}$$

$$S2 = \frac{(\tanh(0.1 \times (\frac{S_{RBM} - \mu_2}{\sigma_2})) + 1)}{2} \tag{17}$$

where $\mu_1$ and $\sigma_1$ refer to the mean and the standard deviation of the scores acquired by the NI. Likewise, $\mu_2$ and $\sigma_2$ represent the mean and the standard deviation of the scores achieved by the RBM. After that, we used average strategy to integrate NI and RBM as follows:

$$S = 0.5 \times S1 + 0.5 \times S2 \tag{18}$$

where $S$ is final predicted SM-miRNA association score matrix.

## RESULTS

### Performance evaluation

We utilized global LOOCV, miRNA-fixed local LOOCV and SM-fixed local LOOCV as well as five-fold cross validation to evaluate the predictive ability of NIRBMSMMA by using two datasets based on SM2miR database (*Liu et al., 2013*). Dataset 1 contains 644 known SM-miRNA associations between 831 SMs and 541 miRNAs. After removing SMs (miRNAs) that have any no known related miRNAs (SMs) in dataset 1, we constructed dataset 2 with 644 known SM-miRNA associations between 39 SMs and 286 miRNAs.

In the LOOCV, each known SM-miRNA association was taken as a test sample in rotation, while the other 633 known SM-miRNA associations were employed as training samples. All unknown SM-miRNA pairs were considered as potential candidate samples. After that, we employed the training samples to train NIRBMSMMA and applied trained NIRBMSMMA to predict scores of test sample and candidate samples. In global LOOCV, we ranked score of test sample according to the scores of all candidate samples. In the SM-fixed LOOCV, based on predicted scores, we sorted test sample with candidate samples that involved fixed SMs. Similarly, in the miRNA-fixed LOOCV, we compared score of test sample with scores of candidate samples that contained fixed miRNAs. Next, the receiver operating characteristics (ROC) curves (AUCs) were plotted by using true positive rate (TPR, sensitivity) as abscissa axis and false positive rate (FPR, 1-specificity) as ordinate axis based on different thresholds. TPR denotes the percentage of candidate samples whose ranking over given threshold, while FPR is the portion of candidate samples with lower rankings than the given threshold. Then, AUCs were computed to compare the performance of NIRBMSMMA with four classic prediction models of TLHNSMMA (*Qu et al., 2018*), SMiR-NBI (*Li et al., 2016*), GISMMA (*Guan et al., 2018*) and SLHGISMMA (*Yin*

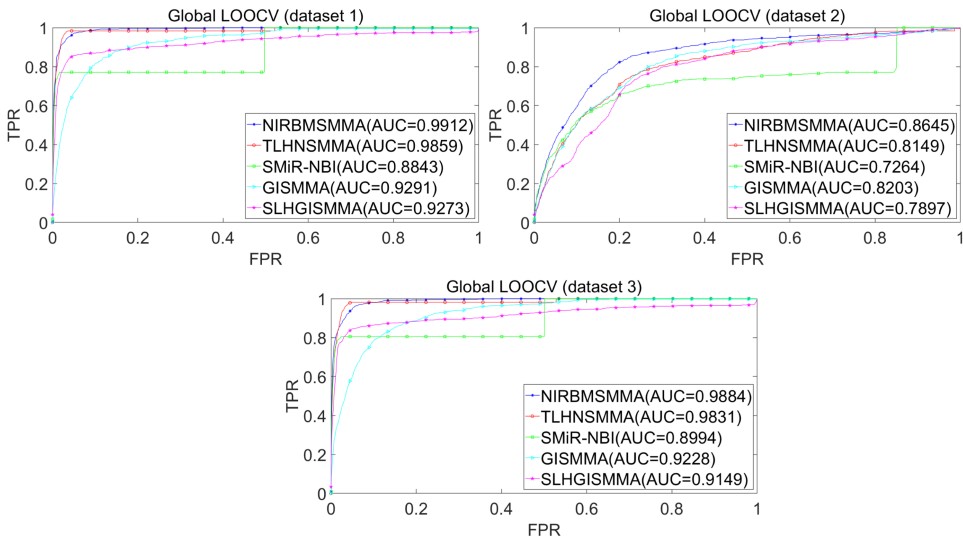

**Figure 3 Performance comparison between NIRBMSMMA and four previous SM-miRNA association models in terms of ROC curves and AUCs based on global and local LOOCV.**

*et al., 2019*). If the value of AUC is equals to 1, it means that NIRBMSMMA has a perfect prediction ability. If the value of AUC is 0.5, it means that prediction of NIRBMSMMA is random. As a result, in global LOOCV, the performance comparisons with some classical algorithms showed that NIRBMSMMA achieved an AUC of 0.9912 and is obviously better than TLHNSMMA (0.9859), SMiR-NBI (0.8843), GISMMA (0.9291), SLHGISMMA (0.9273) based on datasets 1, which was shown in Fig. 3. NIRBMSMMA achieved an AUC of 0.8645 and is obviously better than TLHNSMMA (0.8149), SMiR-NBI (0.7264), GISMMA (0.8203) and SLHGISMMA (0.7897) based on dataset 2 shown in Fig. 3. In the miRNA-fixed local LOOCV, NIRBMSMMA derived an outstanding performance with AUCs of 0.9875, which is better than TLHNSMMA (0.9845), SMiR-NBI (0.8837), GISMMA (0.9505) and SLHGISMMA (0.9553) based on datasets 1 shown in Fig. 4. NIRBMSMMA derived AUC of 0.8720, which is better than TLHNSMMA (0.8244), SMiR-NBI (0.7846), GISMMA (0.8203) and SLHGISMMA (0.8106) based on dataset 2 shown in Fig. 4. In the SM-fixed local LOOCV, NIRBMSMMA gained AUCs of 0.8394, which is better than TLHNSMMA (0.7645), SMiR-NBI (0.7497), GISMMA (0.7702) and SLHGISMMA (0.7702) based on dataset 1 shown in Fig. 5. In the SM-fixed local LOOCV, NIRBMSMMA gained AUCs of 0.7066, the AUCs of other comparison algorithms were TLHNSMMA (0.6057), SMiR-NBI (0.6100), GISMMA (0.6591) and SLHGISMMA (0.6565) based on dataset 2 shown in Fig. 5.

We further employed five-fold cross validation to evaluate the performance of the model. In the five-fold cross validation, 664 known SM-miRNA associations were randomly divided into five portions where four portions contain 133 known SM-miRNA associations, respectively, remaining portion contains 132 known SM-miRNA associations. Each portion was used as test sample in turn and remaining four portions were employed as training samples. Similarly, all unknown SM-miRNA pairs were considered as candidate samples.
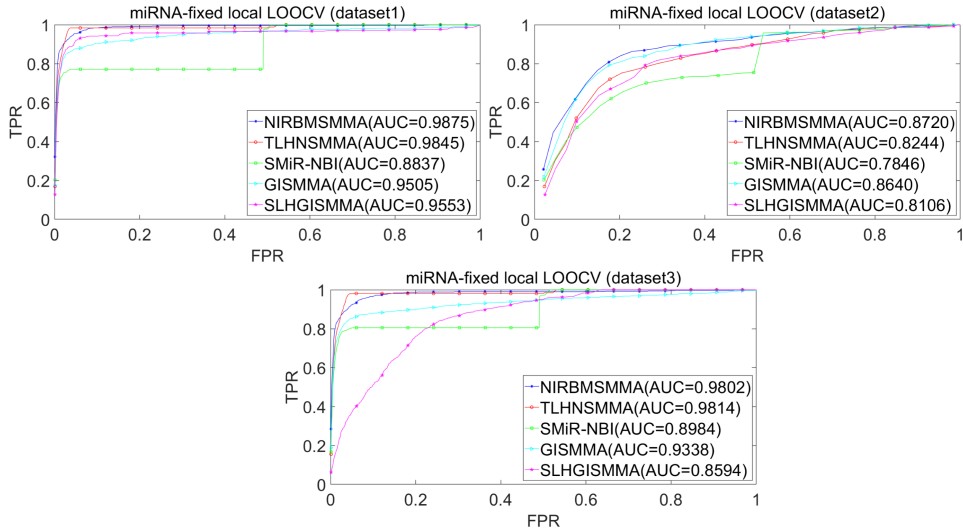

**Figure 4** Performance comparison between NIRBMSMMA and four previous SM-miRNA association models in terms of ROC curves and AUCs based miRNA-fixed LOOCV.

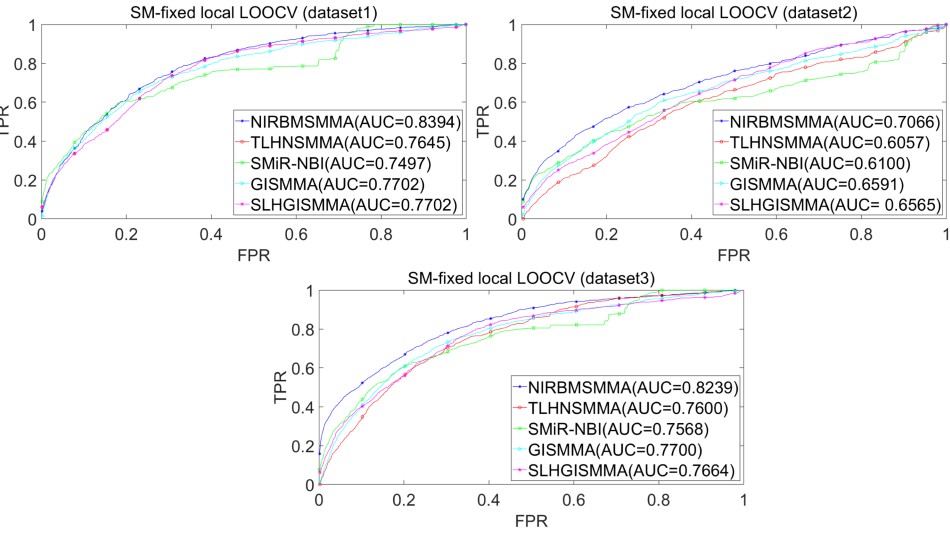

**Figure 5** Performance comparison between NIRBMSMMA and four previous SM-miRNA association models in terms of ROC curves and AUCs based on SM-fixed LOOCV.

Additionally, we utilized the training samples to train NIRBMSMMA and applied trained NIRBMSMMA to predict scores of test sample and candidate samples. It's worth noting that we repeated the five-fold cross-validation process 100 times to avoid bias from the sample division. Consequently, during the five-fold cross-validation process, the AUCs and standard deviations of NIRBMSMMA is $0.9898 \pm 0.0009$, which is better than TLHNSMMA ($0.9851 \pm 0.0012$), SMiR-NBI ($0.8554 \pm 0.0063$), GISMMA ($0.9263 \pm 0.0026$), SLHGISMMA ($0.9446 \pm 0.0036$) based on dataset 1. Simultaneously, the
AUCs and standard deviations of NIRBMSMMA is $0.8547 \pm 0.0046$, which is better than TLHNSMMA ($0.8168 \pm 0.0022$), SMiR-NBI ($0.7104 \pm 0.0087$), GISMMA ($0.8088 \pm 0.0044$), SLHGISMMA ($0.7276 \pm 0.0045$) based on dataset 2.

Dataset 1 and dataset 2 contain the same information of 664 known SM-miRNA associations. The results showed that NIRBMSMMA have better performance in dataset 1 than dataset 2 based on cross validation. The reason is that datasets 1 and dataset 2 have the same positive samples (divided into training and testing samples), but compared to dataset 2, dataset 1 has more candidate samples. Candidate samples in dataset 1 contain more SMs/miRNAs that have any no known associations with miRNAs/SMs. Therefore, the predicted scores of candidate samples in dataset 1 are relatively low compared to the test samples, resulting in higher AUC values based on dataset 1 than dataset 2. Unsurprisingly, adding more known SM-miRNA associations to the dataset 1 is expected to improve the accuracy of NIRBMSMA.

In addition, we further collected 130 known SM-miRNA associations by searching the latest references. Subsequently, the collected dataset was integrated into dataset 1 to build a new dataset of dataset 3. Therefore, dataset 3 contains 794 known SM-miRNA associations between 831 SMs and 531 miRNAs. To evaluate whether the proposed model is applicable to the new dataset, we further implemented LOOCV and 5-fold cross validation based on the new built dataset 3 that was constructed according to dataset 1 and latest references. The results showed that in the global LOOCV, NIRBMSMMA obtained AUC of 0.9884, which is better than TLHNSMMA (0.9831), SMiR-NBI (0.8994), GISMMA (0.9228) and SLHGISMMA (0.9150) shown in Fig. 3. In the miRNA-fixed local LOOCV, NIRBMSMMA and TLHNSMMA derived AUC of 0.9802 0.9814 respectively, which is better than SMiR-NBI (0.8984), GISMMA (0.9338) and SLHGISMMA (0.8594) shown in Fig. 4. In the SM-fixed local LOOCV, NIRBMSMMA gained AUC of 0.8239, which is better than TLHNSMMA (0.7600), SMiR-NBI (0.7568), GISMMA (0.7700) and SLHGISMMA (0.7664) shown in Fig. 5. In the five-fold cross validation, 794 known SM-miRNA associations were randomly divided into five portions where four portions contain 159 known SM-miRNA associations respectively, remaining portion contains 158 known SM-miRNA associations. Similarity, during the five-fold cross-validation process, the AUCs and standard deviations of NIRBMSMMA is $0.9870 \pm 0.0015$, which is better than TLHNSMMA ($0.9818 \pm 0.0016$), SMiR-NBI ($0.8720 \pm 0.0066$), GISMMA ($0.9185 \pm 0034$), SLHGISMMA ($0.8013 \pm 0.1151$).

## Discussing parameters of model

In the NIRBMMMA, there are some parameters, including threshold $\sigma_{upper}$ used in NI, play a crucial role in predictive performance and needed to be determined. According to previous work (*Zhang et al., 2016*), we tested 10 different values of $\sigma_{upper}$ ranging from 0.05 to 0.5 with step 0.05 and computed corresponding 10 AUPR values by employing five-fold cross validation on the training samples. Subsequently, the value of $\sigma_{upper}$ that produced the highest AUPR value was selected and adopted as the optimal parameter to identify new SM-miRNA associations on the basis of test sample.

Likewise, for parameter of number of hidden layer node $s$ used in RBM, we tested 11 candidate values in the range 20 to 120 with step 10, calculated corresponding 11 AUPR values and used $S$ with the best AUPR value to implement prediction. Besides, visible layer node bias $b_i$, hidden layer node bias $c_j$ as well as weight $w_{ij}$ between $i-th$ visible layer node $v_i$ and $j-th$ hidden layer node $h_j$ also play important roles in RBM. We used contrastive divergence (CD) algorithm (*Hinton, 2002*) to determined $b_i$, $c_j$ and $w_{ij}$ based on training temples.

## Case studies

To further evaluate prediction performance of NIRBMSMMA, we implemented two different types of case studies. For the first type of case studies, we used NIRBMSMMA to predict potential association scores for all unknown SM-miRNA pairs. Then, based on predicted scores, we sorted predicted SM-miRNA associations in descending order and confirmed the top 50 predicted SM-miRNA associations by searching the published literature on PubMed. As a result, five (10) out of the top 20 (50) potential SM-miRNA associations were confirmed (see Table 1). For example, the association between gemcitabine and mir-24-2 was predicted and ranked third. An experiment implemented by *Pandita et al. (2015)* indicated that mir-24-2 overexpression cells reduced 37%–50% activity when these cells were treated by using gemcitabine with lower than $IC_{50}$ dose. The association between mir-203a and 5-aza-2′-deoxycytidine ranked the fifth according to predicted scores. *Liu et al. (2016)* found that mir-203a expression increased when using 5-aza-2′-deoxycytidine to treat esophageal cancer cells. The association between 5-Fluorouracil (5-FU) and let-7c ranked the tenth according to predicted scores. *Peng et al. (2015)*found that overexpression of let-7c reduce Akt2 expression and under expression of Akt2 can enhance the sensitivity of renal cell to 5-Fluorouracil.

Also, to assess ability of NIRBMSMMA to predict potential miRNAs for new SM that without any known associated miRNA, based on dataset 1, we conducted the second type of case study on the two investigated SMs of 5-FU and Gemcitabine. First, for an investigated SM, we removed all known associated miRNAs for the investigated SM and used NIRBMSMMA to predict potential miRNAs for the investigated SM. After that, we ranked the top 50 predicted associations between investigated SM and miRNAs according to predicted scores.

5-FU is a kind of chemotherapeutical drug that can prevent cell proliferation by inhibiting enzyme thymidylate synthase and has been used to treat several cancers including head-and-neck (H&N), colorectal and breast (*Goirand et al., 2018*; *Tozer et al., 2019*; *Wigmore et al., 2010*). After implementing NIRBMSMMA, we acquired miRNAs that are potentially associated with 5-FU. Then, we sorted potential associations between 5-FU and miRNAs in descending order and confirmed the top 50 potential associations by using SM2miR database and searching the published literature on PubMed. Particularly, in Tables 2 and 3, we employed PubMed ID 26198104 to denote SM2miR database. As a result, 13 (32) out of the top 20 (50) potential miRNAs correlated with 5-FU were confirmed (see Table 2). For example, the mir-874 was predicted to be associated with 5-FU and the associations ranked the ninth. *Han et al. (2016)* found that mir-874 decrease the resistance of colorectal cancer

**Table 1  Result of the top 50 predicted SM-miRNA associations based on dataset 1.**

| SM | miRNA | Evidence | SM | miRNA | Evidence |
|---|---|---|---|---|---|
| CID:3385 | hsa-mir-1322 | Unconfirmed | CID:60750 | hsa-mir-1-1 | Unconfirmed |
| CID:451668 | hsa-mir-18a | Unconfirmed | CID:3229 | hsa-let-7g | Unconfirmed |
| CID:60750 | hsa-mir-24-2 | 25841339 | CID:451668 | hsa-mir-143 | Unconfirmed |
| CID:451668 | hsa-mir-19b-1 | Unconfirmed | CID:60750 | hsa-let-7a-3 | Unconfirmed |
| CID:451668 | hsa-mir-203a | 26577858 | CID:5311 | hsa-mir-944 | Unconfirmed |
| CID:60750 | hsa-mir-133a-2 | Unconfirmed | CID:60750 | hsa-let-7e | 19654291 |
| CID:60750 | hsa-mir-16-2 | Unconfirmed | CID:60750 | hsa-let-7f-1 | Unconfirmed |
| CID:3385 | hsa-let-7i | Unconfirmed | CID:451668 | hsa-let-7b | 26708866 |
| CID:60750 | hsa-mir-26a-2 | Unconfirmed | CID:448537 | hsa-mir-21 | 28265775 |
| CID:3385 | hsa-let-7c | 25951903 | CID:451668 | hsa-let-7d | 26802971 |
| CID:3385 | hsa-let-7b | 25789066 | CID:3385 | hsa-mir-221 | 27501171 |
| CID:3229 | hsa-mir-29a | Unconfirmed | CID:3385 | hsa-mir-26a-1 | Unconfirmed |
| CID:3385 | hsa-mir-302e | Unconfirmed | CID:451668 | hsa-mir-29c | Unconfirmed |
| CID:5757 | hsa-mir-141 | Unconfirmed | CID:5311 | hsa-mir-1229 | Unconfirmed |
| CID:3385 | hsa-mir-200a | Unconfirmed | CID:3229 | hsa-mir-27a | Unconfirmed |
| CID:60750 | hsa-mir-199a-1 | Unconfirmed | CID:3385 | hsa-mir-194-1 | Unconfirmed |
| CID:5311 | hsa-mir-30c-2 | Unconfirmed | CID:60750 | hsa-mir-194-2 | Unconfirmed |
| CID:3385 | hsa-mir-126 | 26062749 | CID:451668 | hsa-mir-106a | Unconfirmed |
| CID:448537 | hsa-mir-92a-2 | Unconfirmed | CID:3385 | hsa-mir-324 | Unconfirmed |
| CID:60750 | hsa-mir-23a | Unconfirmed | CID:451668 | hsa-let-7a-1 | Unconfirmed |
| CID:448537 | hsa-mir-181a-2 | Unconfirmed | CID:448537 | hsa-mir-605 | Unconfirmed |
| CID:60750 | hsa-let-7a-2 | Unconfirmed | CID:3385 | hsa-mir-19b-1 | Unconfirmed |
| CID:5311 | hsa-mir-24-2 | Unconfirmed | CID:448537 | hsa-mir-662 | Unconfirmed |
| CID:448537 | hsa-mir-27a | Unconfirmed | CID:60750 | hsa-let-7g | Unconfirmed |
| CID:3385 | hsa-mir-181a-1 | Unconfirmed | CID:451668 | hsa-mir-132 | Unconfirmed |

cells to 5-FU and inhibit proliferation of colorectal cancer cells. The mir-455 was predicted to associate with 5-FU and the association ranked the tenth. An experiment conducted by *Hummel et al. (2011)* showed that the expression level of mir-455-3p was decreased after treatment of 5-FU by using TargetScan and bioinformatic analysis. The associations between mir-299 and 5-FU ranked the thirty-fourth according to predicted score. *Chen, Lu & Hu (2019)* found that mir-299 overexpression can increase sensibility of Hexokinase 1 (HK1) cells to 5-FU and further decrease invasion ability of HK1 cells.

Gemcitabine, a pyrimidine nucleoside analog anticancer drug, has potent activity for a wide spectrum of solid tumors (*Miao, Chen & Luan, 2020*; *Mini et al., 2006*; *Rizzuto, Ghazaly & Peters, 2017*). After implemented NIRBMSMMA, we gained potential Gemcitabine-miRNA associations score. After that, we sorted potential Gemcitabine-miRNA associations in descending order and confirmed the top 50 potential associations by searching the published literature on PubMed. Result showed that 11 (32) out of the top 20 (50) potential miRNAs associated with Gemcitabine were confirmed (see Table 3). For example, the mir-17 was predicted to be associated with gemcitabine and the association ranked the twelfth. *Yan et al. (2012)* found that mir-17-5p inhibitor can enhance sensitivity
Table 2 **Result of the top 50 predicted miRNAs associated with 5-fluorouracil based on dataset 1.**

| miRNA | Evidence | miRNA | Evidence |
|---|---|---|---|
| hsa-mir-324 | Unconfirmed | hsa-let-7c | 25951903 |
| hsa-mir-24-1 | 26198104 | hsa-mir-320a | 26198104 |
| hsa-mir-23a | 26198104 | hsa-mir-345 | Unconfirmed |
| hsa-mir-501 | 26198104 | hsa-mir-155 | 28347920 |
| hsa-mir-1226 | 26198104 | hsa-mir-194-1 | Unconfirmed |
| hsa-mir-24-2 | 26198104 | hsa-mir-337 | Unconfirmed |
| hsa-mir-650 | Unconfirmed | hsa-mir-212 | Unconfirmed |
| hsa-mir-500a | Unconfirmed | hsa-mir-205 | 24396484 |
| hsa-mir-874 | 27221209 | hsa-mir-299 | 31786874 |
| hsa-mir-455 | 21743970 | hsa-mir-128-2 | 26198104 |
| hsa-mir-27a | 26198104 | hsa-mir-197 | 26198104 |
| hsa-mir-21 | 26198104 | hsa-mir-128-1 | 26198104 |
| hsa-mir-181a-1 | 29795190 | hsa-mir-199a-2 | 26198104 |
| hsa-let-7a-1 | 26198104 | hsa-mir-132 | 26198104 |
| hsa-mir-328 | Unconfirmed | hsa-mir-373 | 32426273 |
| hsa-let-7b | 25789066 | hsa-mir-139 | 27173050 |
| hsa-mir-181a-2 | 24462870 | hsa-mir-129-2 | Unconfirmed |
| hsa-mir-26a-1 | Unconfirmed | hsa-mir-1-2 | Unconfirmed |
| hsa-mir-346 | Unconfirmed | hsa-mir-329-1 | Unconfirmed |
| hsa-mir-181b-1 | unconfirmed | hsa-mir-149 | 26198104 |
| hsa-let-7d | 26198104 | hsa-mir-342 | 26198104 |
| hsa-mir-181b-2 | unconfirmed | hsa-mir-211 | 28720546 |
| hsa-mir-16-1 | 26198104 | hsa-mir-187 | Unconfirmed |
| hsa-mir-133a-1 | 26198104 | hsa-mir-329-2 | Unconfirmed |
| hsa-mir-409 | Unconfirmed | hsa-mir-326 | 26239225 |

of pancreatic cancer cells to gemcitabine by upregulating Bim protein expression. Mir-27b was also predicted to associate with gemcitabine. A study implemented by *Bera et al. (2014)* demonstrated that the expression level of mir-27b is significantly decreased in gemcitabine resistant pancreatic ductal adenocarcinoma cells by testing with quantitative polymerase chain reaction (qPCR) analysis. Moreover, the mir-125a was predicted to be associated with gemcitabine and the association ranked twenty-third. *Yao et al. (2016)* found that mir-125a can enhance chemo-resistance of pancreatic cancer cells to gemcitabine *via* targeting A20 gene.

Our main research interest is in computational bioinformatics. Therefore, we usually confirmed the predicted results presented in case study by databases and published literatures. For some predicted association information that is not validated by any study, we hope the predicted associations can be further confirmed by biologist based on biological experiments in the future.

**Table 3  Result of the top 50 predicted miRNAs associated with gemcitabine based on dataset 1.**

| miRNA | Evidence | miRNA | Evidence |
|---|---|---|---|
| hsa-mir-24-1 | 26198104 | hsa-mir-320a | 26198104 |
| hsa-mir-24-2 | 25841339 | hsa-mir-23b | Unconfirmed |
| hsa-mir-23a | Unconfirmed | hsa-mir-18a | Unconfirmed |
| hsa-mir-501 | Unconfirmed | hsa-mir-93 | 34162560 |
| hsa-mir-1226 | Unconfirmed | hsa-mir-25 | 24040438 |
| hsa-mir-324 | 26198104 | hsa-mir-15a | 31846800 |
| hsa-mir-874 | Unconfirmed | hsa-mir-182 | 25833690 |
| hsa-mir-650 | Unconfirmed | hsa-mir-191 | Unconfirmed |
| hsa-mir-500a | Unconfirmed | hsa-let-7f-1 | Unconfirmed |
| hsa-mir-455 | Unconfirmed | hsa-mir-106b | 31374207 |
| hsa-mir-19a | 26041879 | hsa-mir-210 | 31713003 |
| hsa-mir-17 | 23001407 | hsa-mir-203a | Unconfirmed |
| hsa-mir-20a | 24924176 | hsa-let-7a-1 | 26198104 |
| hsa-mir-125b-1 | Unconfirmed | hsa-let-7a-3 | Unconfirmed |
| hsa-mir-125b-2 | Unconfirmed | hsa-mir-663a | 30788003 |
| hsa-mir-27b | 25184537 | hsa-mir-145 | 28839463 |
| hsa-mir-21 | 26198104 | hsa-let-7a-2 | Unconfirmed |
| hsa-mir-27a | 26198104 | hsa-mir-200c | 19654291 |
| hsa-mir-638 | 23293055 | hsa-mir-222 | 28743280 |
| hsa-mir-31 | 26606261 | hsa-let-7e | 19654291 |
| hsa-mir-16-2 | Unconfirmed | hsa-mir-200b | 30628651 |
| hsa-mir-16-1 | 26198104 | hsa-mir-141 | 33468723 |
| hsa-mir-125a | 26758190 | hsa-mir-30a | 30770779 |
| hsa-let-7g | Unconfirmed | hsa-mir-15b | 26166038 |
| hsa-mir-29a | 26198104 | hsa-mir-128-1 | 26198104 |

## DISCUSSION

Traditional wet experiment methods are time-consuming and labor-intensive in identifying potential SM-miRNAs associations. The computational approaches can overcome disadvantages above. In this article, we developed a computational model of NIRBMSMMA to identify potential SM-MiRNA associations by employing ensemble learning to integrate NI and RBM. Moreover, LOOCV, five-fold cross validation, and two types of case studies indicated that NIRBMSMMA has outstanding prediction ability and stable prediction performance.

The effectiveness of NIRBMSMMA mainly comes from the following factors. First, the NIRBMSMMA integrated two methods of NI and RBM. NI can fully utilize similarity information to filtrate neighbors of SM or miRNA with different thresholds for predicting potential SM-miRNA associations, while RBM can learn hidden probability distribution from known SM-miRNA associations to identify new SM-miRNA associations. Second, ensemble learning was used to integrate NI and RBM to overcome the problem of poor normalization ability of individual predictor for obtaining better predicted performance. Third, in the NIRBMSMMA, we used multiple highly reliable biological data including

known SM-miRNA associations, four different types of SM similarity data and two different types of miRNA similarity data, which can improve the prediction accuracy of the model compared with using a kind of similarity data.

However, there are also some drawbacks for NIRBMSMMA. First, the known SM-miRNA associations are few compared to the unknown SM-miRNA pairs, which limits the predictive performance of our model. Second, it is possible that the two base predictors utilized in the ensemble learning may not be sufficient, and additional, more dependable base predictors may lead to even better performance of the NIRBMSMMA.

## CONCLUSIONS

In this article, a novel computational model of NIRBMSMMA was proposed to identify potential small molecule-MiRNA associations based on known SM-miRNA associations, SM similarity and miRNA similarity by integrating NI and RBM. We also implemented four different types of cross validation and case studies to evaluate the performance of NIRBMSMMA. The results demonstrated that the performance of NIRBMSMMA is efficient in the identification of potential molecule-MiRNA associations prediction.

### Funding

This research was supported by the Natural Science Foundation of Jiangsu Province under Grant BK20220621, the Natural Science Fund Project of Colleges in Jiangsu Province 21KJB520030, the National Natural Science Foundation of China under Grant 62101645, 62206177, the Zhejiang Provincial Natural Science Foundation of China under Grant LQ22F020024, LY23F020007, LTY22F020003, and the Postgraduate Research & Practice Innovation Program of Jiangsu Province:KYCX22_3065. The funders had no role in study design, data collection and analysis, decision to publish, or preparation of the manuscript.

### Grant Disclosures

The following grant information was disclosed by the authors:
Natural Science Foundation of Jiangsu Province: BK20220621.
Natural Science Fund Project of Colleges in Jiangsu Province: 21KJB520030.
National Natural Science Foundation of China: 62101645, 62206177.
Zhejiang Provincial Natural Science Foundation of China: LQ22F020024, LY23F020007, LTY22F020003.
Postgraduate Research & Practice Innovation Program of Jiangsu Province: KYCX22_3065.

### Competing Interests

The authors declare there are no competing interests.

### Author Contributions

- Jia Qu conceived and designed the experiments, performed the experiments, authored or reviewed drafts of the article, and approved the final draft.

- Zihao Song conceived and designed the experiments, performed the experiments, prepared figures and/or tables, and approved the final draft.
- Xiaolong Cheng conceived and designed the experiments, performed the experiments, prepared figures and/or tables, and approved the final draft.
- Zhibin Jiang analyzed the data, authored or reviewed drafts of the article, and approved the final draft.
- Jie Zhou analyzed the data, authored or reviewed drafts of the article, and approved the final draft.

## Data Availability

Code and raw data are available in the Supplemental Files.

## Supplemental Information

Supplemental information for this article can be found online at http://dx.doi.org/10.7717/peerj.15889#supplemental-information.

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
