# Peer review of "Neighborhood-based inference and restricted Boltzmann machine for small molecule-miRNA associations prediction"

_PeerJ, doi:10.7717/peerj.15889_

## Round 0.1 · original submission · Major Revisions

Your manuscript has been reviewed by three experts in the field. As you can see from their comments below, one of them recommends its acceptance basically while the others point out a few points to be improved. Please read their comments carefully and revise the manuscript accordingly. Particularly, I agree with Reviewer 1's opinion that the authors should try to apply their method to new data; using only the same old data with past methods would be convenient for comparing their performance but it gives the readers the impression that this work is less innovative. Looking forward to the resubmission of the revised manuscript.

Reviewer 1 ·

Basic reporting

no comment

Experimental design

1. In Method section, the authors utilized two SM-miRNA dataset, dataset1 (831 SMs and 531 miRNAs) and dataset2 (39 SMs and 286 miRNAs). Based on my knowledge, the RFSMMA (PMID: 30840454) and SNMFSMMA (PMID: 31739716) also used the same dataset 1 and 2 in their analysis. Are there any new dataset published dataset during these years? These two datasets may be out of date for a newly designed tool. It can be beneficial to include more up-to-date datasets to ensure the applicability of the newly designed tool.

Validity of the findings

1. The authors should claim more about the motivation to develop a new tool. What are the limitations of previous tool? What gap did the authors fill in this field? The authors should discuss more in the introduction section.

Additional comments

1. In the last paragraph of introduction, the authors don’t need to repeat all the results here.

Reviewer 2 ·

Basic reporting

no comment

Experimental design

no comment

Validity of the findings

no comment

Additional comments

MicroRNAs have be considered as a type of novel small molecule drug targets for disease treatment. However, wet-experimental approaches are time-consuming and labor-intensive for identifying small molecule-microRNA association. Hence, authors proposed a novel computational method to identify potential association between small molecule and microRNA by using neighborhood-based inference and restricted Boltzmann machine based on known associations, small molecule similarity and miRNA similarity. A series of experiments and comparisons with literature methods have demonstrated the performance of the current method. However, the following issues need to be solved.
1. In this manuscript, various similarity information of small molecules and microRNAs is used to predict potential associations between them. However, the current method has not been demonstrated for its performance in similarity. For example, dataset 1 and dataset 2 contain 664 known small molecule-microRNA associations. what will happen to the performance of the model if redundant associations are removed.
2. Line 191, after eliminating SM-miRNA pairs of unknown in dataset 1? How man unknown association pairs are there? How do you know they are unknown? Why delete these unknown associations?
3. Line 258, potential microbe-drug associations score?
4. Line 419, IC50, in which 50 should be located in the lower corner.
5. Finally, authors identified some potential associations between small molecule and microRNA. These potential associations can be further validated through experiments, such as molecular docking simulations.

Reviewer 3 ·

Basic reporting

The English expression of this article is basically clear, unambiguous, and technicallly corrcet. The introduction section of the article provides a comprehensive discussion, analyzing and introducing some representative research achievements. The structure of the article and the figures and tables in the article basically comply with professional standards. Finally, the paper meets the requirement of self-contained.

There are also a few syntax error in the paper, for example, the first letter of “prediction” in the title of the article should be capitalized. Or all words with lowercase initial letters?

Experimental design

The experimental design of the article is reasonable. Methods described in the article provides sufficient detail to replicate.

Validity of the findings

The conclusions of the article are clear and well stated, providing raw data and code, and the experimental results support the relevant conclusions.

---

## Round 0.2 · Minor Revisions

Although all of the reviewers now agree to accept the revised manuscript, in principle, Reviewer 1 gives a technical point to be improved, which I agree with. Could you re-revise the manuscript (figure) accordingly? Looking forward to your final version.

Reviewer 1 ·

Basic reporting

no comment

Experimental design

no comment

Validity of the findings

no comment

Additional comments

Please enhance the quality of Figures 3, 4, and 5 by increasing the font size for the titles and the numbers on the X and Y axes. The current font size is too small and may make it difficult for readers.

Reviewer 2 ·

Basic reporting

no comment

Experimental design

no comment

Validity of the findings

no comment

Additional comments

no comment

---

## Round 0.3 · accepted · Accept

I confirm that the authors addressed the remaining point and thus I am happy to recommend its acceptance to the section editor. Congratulations!